# Numerical Investigation on the Effects of Grain Size and Grinding Depth on Nano-Grinding of Cadmium Telluride Using Molecular Dynamics Simulation

**DOI:** 10.3390/nano13192670

**Published:** 2023-09-29

**Authors:** Changlin Liu, Wai Sze Yip, Suet To, Bolong Chen, Jianfeng Xu

**Affiliations:** 1State Key Laboratory of Ultra-Recision Machining Technology, The Hong Kong Polytechnic University, Hong Kong 999077, China; changlin.liu@polyu.edu.hk (C.L.); lenny.ws.yip@polyu.edu.hk (W.S.Y.); 2The Hong Kong Polytechnic University Shenzhen Research Institute, Shenzhen 518000, China; 3State Key Laboratory of Intelligent Manufacturing Equipment and Technology, Huazhong University of Science and Technology, Wuhan 430074, China; m202270535@hust.edu.cn

**Keywords:** cadmium telluride, nano-grinding, molecular dynamics simulation, material removal mechanism, subsurface damage

## Abstract

Cadmium telluride (CdTe) is known as an important semiconductor material with favorable physical properties. However, as a soft-brittle material, the fabrication of high-quality surfaces on CdTe is quite challenging. To improve the fundamental understanding of the nanoscale deformation mechanisms of CdTe, in this paper, MD simulation was performed to explore the nano-grinding process of CdTe with consideration of the effects of grain size and grinding depth. The simulation results indicate that during nano-grinding, the dominant grinding mechanism could switch from elastic deformation to ploughing, and then cutting as the grinding depth increases. It was observed that the critical relative grain sharpness (*RGS*) for the transition from ploughing to cutting is greatly influenced by the grain size. Furthermore, as the grinding depth increases, the dominant subsurface damage mechanism could switch from surface friction into slip motion along the <110> directions. Meanwhile, as the grain size increases, less friction-induced damage is generated in the subsurface workpiece, and more dislocations are formed near the machined groove. Moreover, regardless of the grain size, it was observed that the generation of dislocation is more apparent as the dominant grinding mechanism becomes ploughing and cutting.

## 1. Introduction

Cadmium telluride (CdTe), known as an important semiconductor material with favorable properties, has drawn extensive attention in the fields of solar cells, medical imaging, and radiation detection [1,2,3]. As a soft-brittle material, the fabrication of high-quality surfaces on CdTe is challenging. Surface and subsurface damage including distorted atoms, dislocation loops, and micro-cracks are usually inevitable during machining [4,5], which can deteriorate the operating performance of CdTe-based devices [6,7]. In recent years, nanoscale machining techniques (NMT) such as nano-scratching, nano-cutting, and nano-grinding have been recognized as effective methods for manufacturing nanoscale surfaces with low subsurface damage. In the nanoscale machining process, as the material removal thickness usually ranges from tens to hundreds of nanometers, the deformation mechanism of workpiece material can be greatly different from macroscale machining. Therefore, a comprehensive understanding of the mechanism in nanoscale machining is essential for high-quality manufacturing of CdTe.

Due to the inaccessibility of the interactions between materials, it is difficult to observe and measure the physical process in nanoscale machining via experiments. At present, molecular dynamics (MD) simulation has merged as a powerful method to study the deformation and fracture mechanisms at a nanoscale [8,9,10,11]. In MD simulation, the evolution of structure including phase transition [12,13], dislocation behavior [14,15], and cracks propagation [5,16] can be investigated at an atomic level. It has been successfully used to research the nanoscale machining mechanisms of various materials. For instance, Ye et al. [17] used MD simulation to investigate the frictional forces, material removal mechanism, and defects of Cu in nanometric cutting. Fang et al. [18] studied the cutting mechanism of single-crystal Si at the nanoscale using MD simulation. They proposed that extrusion is the dominant mechanism in the ductile removal of single-crystal Si when the material removal thickness is reduced to the nanoscale. Goel et al. [19] used MD to investigate the crystal anisotropy effect of 3C-SiC on the material removal mechanism, cutting forces, and internal stress during nanometric cutting. Xiao et al. [20] conducted an MD simulation to study the brittle–ductile cutting mode transition of 6H-SiC by utilizing GPU computation and adopting the Vashishta potential. Lai et al. [21,22] conducted an MD simulation of nanoscale cutting on single-crystal Ge to study the anisotropic characteristics in subsurface deformation and side flow of Ge atoms with different machining parameters. Fan et al. [23] performed an MD simulation of nanometric cutting on Ni-Fe-Cr alloy. They analyzed the dislocation behavior in the workpiece and revealed the work hardening mechanism. Avila et al. [24] used the MD method to explore the effect of the tool rake angle on the nanoscale cutting of CuZr metallic glass. They analyzed the material removal behavior and proposed that the formation of parallel shear bands governs the plasticity of metallic glass during machining. Lin et al. [25] conducted a nano-scratching simulation of the C-plane sapphire using the MD method and discussed the material removal mechanism and subsurface damage formation. Papanikolaou and Salonitis [26] explored the effect of grain size on the nanometric cutting mechanism of pure Al using MD simulation. They built a polycrystalline workpiece with different grain sizes using the melt-quench method with different cooling rates. Fan et al. [27] studied the wear mechanism of diamonds during the tip-based nano-scratching of GaAs using MD simulation. They investigated the elastic–plastic deformation mechanism and the *sp*^3^–*sp*^2^ transition of the diamond tip. These studies revealed the nanoscale machining mechanism for a variety of materials and provided strategies to analyze the material removal behavior and subsurface damage in nanoscale machining.

In the 1950s, Loferski first outlined a wide range of applications of CdTe in the photovoltaic field [28]. Then, several papers revealed the existence of dislocations in CdTe during deformation, which mainly focus on the effects of dislocations on its optical and electrical properties [29,30,31]. Zhang et al. [32] conducted MD simulations of nano-indentation of CdTe to reveal the deformation mechanism and hardening effect of twin boundaries (TBs). They also investigated the deformation and crack mechanisms of monocrystalline and nano-twinned CdTe [33]. Xiang et al. [34] studied the effect of in-plane anisotropy and TBs on the softening and strengthening of CdTe via MD simulation of the nano-indentation process. They found that the indenter direction affects the plastic deformation and dislocation behavior of CdTe films, causing different softening/hardening effects. However, to the best of the authors’ knowledge, no attempts have been made to explore the nanoscale material removal mechanism of CdTe during nanoscale machining, which is critical for improving the theory of the high-quality surface fabrication of CdTe components. Therefore, in the present work, MD simulation was performed to investigate the nano-grinding mechanism of CdTe. The effects of the grinding depth and grain size on the material removal and subsurface damage formation mechanism were studied. The simulation was conducted using the Large-scale Atomic/Molecular Massively Parallel Simulator (LAMMPS) [35], and the simulation results were analyzed by OVITO [36].

## 2. Simulation Methods

Figure 1 presents the MD model for nano-grinding simulation. The CdTe workpiece is set as deformable while a single abrasive grain is modeled as a repulsive sphere by the “fix indent” command from LAMMPS. It inserts a spherical indenter within a simulation box [37,38,39]. The interaction between workpiece atoms and abrasive grain is described by:(1)$$V(r)=\{\begin{array}{c} -k\;{\left(r\;-\;R\right)}^{2},\;r\;\leq \;R\\ \;0,\;\;\;r\;>\;R \end{array}$$

Here, the parameter *R* represents the radius of the abrasive grain while *k* is the specified force constant and represents the stiffness of the grain. It is believed that the precise choice of *k* is unimportant in a similar system [40]. In this simulation, *k* is set as 5 eV/A^3^ referring to previous simulations [41,42].

Adopting an appropriate potential is critical for the accuracy of the results in MD simulation. In this paper, the Stillinger–Weber potential is applied to describe the interactions between workpiece atoms [43,44]. It has been widely used to investigate the nanoscale deformation mechanism of the CdTe/CdS system [45,46]. The periodic boundary condition is set for the *x* and *y* directions to eliminate the boundary effect of the workpiece. As shown in Figure 1, the CdTe workpiece is classified into Newtonian region, boundary region, and thermostat region. Before machining, the simulation system is relaxed by applying the isothermal–isobaric (NPT) ensemble. During the nano-grinding process, atoms in the Newton region follow Newton’s law of motion by applying the microcanonical (NVE) ensemble. While the atoms in the thermostat region are kept at the ambient temperature in the canonical (NVT) ensemble to dissipate the generated heat, atoms in the boundary region are fixed in their balanced positions to hold the workpiece. The nano-grinding simulation is conducted with increasing material removal thickness and various abrasive radii. Each simulation case is repeated three times to eliminate the statistical errors. Simulation details are listed in Table 1.

## 3. Results and Discussion

### 3.1. Material Removal Mechanism

Figure 2 shows the morphology of the machined surface at different stages during nano-grinding. It can be concluded that the material removal behavior of CdTe is significantly influenced by grinding depth and grain size. When the grinding depth is small, the workpiece mainly experiences elastic deformation and no atoms are extruded upon the uncut surface. The friction between the abrasive grain and the workpiece could cause the near-surface structural change of the workpiece. When the grinding depth is increased, plastic deformation occurs and the machining groove gradually becomes apparent, indicating that the grinding mechanism becomes ploughing [47]. For smaller grains, some of the workpiece atoms are extruded to the front and side region of the abrasive grain to form chips and ridges while more atoms are compressed into the workpiece when large grains are used. As the grinding depth further increases, chips and ridges are formed continuously on the uncut surface, indicating that the material removal behavior becomes a stable cutting process. Figure 3 shows the machined surface morphologies after grinding where the atoms in chips and ridges are hidden for a clear picture of the machined groove. The elastic deformation zone can be observed clearly at the initial grinding stage where the workpiece material is fully recovered after the abrasive grain passes. When grains with larger size is adopted, penetration of the abrasive grains on the workpiece surface can be suppressed and a more elastic deformation zone is observed, indicating that the critical grinding depth for plastic deformation is increased.

Variation of the atoms in chips and ridges as a function of grinding depth is present in Figure 4a. In the initial stage, more atoms are piled up on the uncut surface with a small grain size since the material removal mechanism transition occurs at a smaller grinding depth. On the other hand, due to the larger contact area between the workpiece and abrasive grain, the atoms in chips and ridges rise at a faster rate with larger grain sizes. It is observed that when the grinding depth reaches 6 nm, most atoms are piled up to the uncut surface as the radius of the grains is set as 8 nm. In nanometric cutting, the critical depth for the transition of the material removal mechanism is greatly determined by the geometric relationship between the depth of cut and tool edge radius, which can be measured by the relative tool sharpness (*RTS*) [48,49]. In this research, the ratio of the transient grinding depth and grain radius is defined as the relative grain sharpness (*RGS*) to describe the geometric relationship between grinding depth and grain size. Figure 4b shows the variation of the critical *RGS* for grinding mechanism transition as a function of grain size, which is determined based on the surface morphology in Figure 3 and piling-up atoms in Figure 4a. With an increase in grinding depth, the dominant grinding mechanism experiences the transition from elastic deformation to ploughing and then cutting. It is observed that critical *RGS* for the transition from elastic deformation to ploughing decreases from 0.23 to 0.16 with an increase in grain size. By contrast, the transition from ploughing to cutting can be affected more apparently by increasing the grain size. As the grain radius increases from 4 nm to 12 nm, the critical *RGS* for ploughing to cutting transition decreases from 0.45 to 0.25. It is worth mentioning that in previous research on nanometric cutting, the critical *RTS* for the transition of material removal mechanism is nearly constant for certain materials and cutting directions [48]. This difference can be attributed to the three-dimensional model adopted in this simulation. Compared to two-dimensional models, the MD model in this simulation could describe the three-dimensional structure evolution and atomic flow along the *y* direction more accurately [50], which is an important phenomenon in the formation of ridges and affects the atomic strain and internal stress near the contact region in the workpiece.

### 3.2. Subsurface Damage

Furthermore, to understand the deformation mechanism in subsurface workpiece during nano-grinding, the atomic displacement magnitudes of workpiece atoms with different grain sizes are calculated, as illustrated in Figure 5a. When elastic deformation occurs, the workpiece atoms deviate from their balance position and the displacement magnitude of their neighbors would vary continuously. During plastic deformation, the crystal structure is destructed and an obvious discontinuous distribution of the displacement magnitude can be observed. It is found that in the initial grinding stage, the friction between the workpiece and abrasive grain could induce irregular destruction of the crystal structure near the workpiece surface. As grinding depth increases, the plastic deformation zone is gradually enlarged and obvious dislocation paths along the <101> directions are observed in the subsurface workpiece. When the grain size increases, both the elastic and plastic deformation zone is apparently enlarged, causing greater displacement of workpiece atoms, as shown in Figure 5b. Although the contact region between the workpiece and abrasive grain is enlarged, the friction-induced damage becomes less apparent when a larger grain is adopted. More dislocation paths are observed as the grain size increases due to the stronger interaction between abrasive grain and workpiece subsurface, as shown in Figure 5a.

In the zincblende structured CdTe, slip along the <110>{111} directions is the dominant slip motion and the predominant dislocation style is the perfect dislocation with Burgers vector 1/2<110>. Figure 6a–c shows the snapshot of the dislocations in the subsurface workpiece during nano-grinding with a grain radius of 10 nm, which is determined by the Dislocation Extraction Algorithm (DXA) [51]. In the initial grinding stage, only some disordered atom clusters with few dislocations are observed due to the friction between the tool and the workpiece. As the grinding depth increases, more dislocations are formed near the machined groove and several dislocation loops are emitted into the subsurface along the <110> direction. Variation of the total dislocation length during nano-grinding is shown in Figure 6d. It is obvious that after the initial frictional interaction, more dislocations are generated in the subsurface workpiece with an increase in the grain size. The generation of the dislocation is greatly affected by the material removal mechanism. To eliminate the size effect of the deformation region under different grains on dislocation statistics, the ratio of the dislocation length during and after nano-grinding as a function of *RGS* was calculated for each case, as shown in Figure 7a. It is observed that regardless of the grain size, the generation of dislocation becomes apparent as the *RGS* rises to about 0.3, where the material is mainly removed through ploughing. Meanwhile, in addition to the perfect dislocation, Shockley partial dislocation with 1/6<112> Burger’s vector can be observed during the grinding process, which can lead to the presence of stacking faults in the subsurface workpiece. With the increase in the grain size, the proportion of the perfect dislocation increases obviously, as shown in Figure 7b.

### 3.3. Grinding Force and Stress Analysis

During the nano-grinding process, grinding forces gradually rise as an increase in grinding depth. Figure 8a presents the transient normal force (*F_z_*) and tangential force (*F_x_*) when the grain radius is 6 nm. In the initial stage of machining, as the workpiece experiences elastic deformation and no material is removed, the normal force gradually rises while the tangential force is nearly zero. As the simulation time reaches about 400 ps, the workpiece material is piled up to form chips and ridges, and the tangential force begins to increase. As the strain energy is dissipated by the plastic deformation in the workpiece, the increase in the normal force with increasing grinding depth is decelerated. Meanwhile, the fluctuation of the grinding forces becomes more apparent due to the accumulation and dissipation of the strain energy in the deformation region. Figure 8b shows the average grinding forces with different grain radii. It is observed that the grinding forces gradually rise as the grain radius increases. The increment of the normal force is more obvious as the contact area between the workpiece and abrasive grain is enlarged with increasing grain size. While the increase in tangential force is less noticeable as the formation of chips and ridges is suppressed with increasing grain size at the same grinding depth, resulting in a decrease in material load along the *x* direction during the nano-grinding process. Furthermore, the average frictional coefficient (*µ*) under different grain sizes is present in Figure 9a, which is calculated by *µ* = *F_x_*/*F_z_*. It is observed that the frictional coefficient is increased as the grain size decreases due to the variation of the dominant grinding mechanism [52]. Moreover, to evaluate the grain size effect on the grindability of CdTe, the specific grinding energy is applied to determine the grinding efficiency. It is defined as the energy required per unit volume of the removed material during grinding. In this simulation, the specific grinding energy is obtained by dividing the input energy by the volume of the removed workpiece materials, including chips and ridges. The variation of the specific grinding energy with different grain sizes is shown in Figure 9b. It is observed that the specific grinding energy is gradually increased as the grain size rises, which indicates a lower grinding efficiency and less energy-efficient grindability of CdTe. This variation can be attributed to the influence of the grain size on the grinding mechanism that elastic deformation and ploughing contribute less to the material removal.

During nano-grinding, the strong interaction between the workpiece and abrasive grain could lead to an increase in the stress in the deformation region, which has a great influence on the grinding process and damage formation. The hydrostatic stress and von Mises stress can be calculated via the three-dimensional stress tensors in LAMMPS [53]:(2)$$\sigma_{\mathrm{hydrostatic}}=\frac{\left(\sigma_{x}+\sigma_{y}+\sigma_{z}\right)}{3}$$
(3)$$\sigma_{\mathrm{von}\;\mathrm{Mises}}=\sqrt{\frac{(\sigma_{x}\;-\;\sigma_{y}{)}^{2}+(\sigma_{y}\;-\;\sigma_{z}{)}^{2}+(\sigma_{z}\;-\;\sigma_{x}{)}^{2}+6(\tau_{\mathrm{xy}}+\tau_{\mathrm{yz}}+\tau_{\mathrm{zx}}{)}^{2}\;}{2}}$$
where *σ_x_*, *σ_y_*, *σ_z_*, *τ_xy_*, *τ_xz_*, and *τ_yz_* represent the stress tensors in the Cartesian coordinate system. Figure 10 shows the distribution of the hydrostatic stress and von Mises stress in the workpiece at a grinding depth of 5 nm when the grain radius is 10 nm. It is observed that a high-compressive region is formed near the grain edge during nano-grinding while shear stress is apparent in the workpiece along the slip motion direction. During the nano-grinding process, the average compressive stress and von Mises stress in the deformation zone rise apparently with the advance of the abrasive grain, as shown in Figure 11a,b. Meanwhile, as the dominant material removal mechanism switches to cutting, the deformation-induced strain energy is periodically accumulated and dissipated by the shearing process, causing obvious fluctuation of the internal stress. Furthermore, as the grain size is increased, the internal stress raises apparently since the contact area between the workpiece and abrasive grain is enlarged, which causes a larger deformation zone in the workpiece. This variation is one of the reasons for the increase in the proportion of perfect dislocation because the crystal deforms primarily by the motion of shuffle dislocations at low temperatures and high stresses [54]. Furthermore, obvious residual stress can be observed on the machined surface, as shown in Figure 10. The residual stress can be caused by the machining-induced plastic deformation and structural changes. Figure 11c, d shows the average residual stress in the machined workpiece. Since stronger deformation is introduced into the workpiece, larger compressive stress and von Mises stress can be observed on the machined surface when the grain size is enlarged.

## 4. Conclusions

In this research, MD simulation was conducted to investigate the nano-grinding process of CdTe. The effects of grinding depth and grain size on the material removal mechanism and grinding damage formation were discussed. The main conclusions are drawn as follows:

(1) During the nano-grinding process, the dominant grinding mechanism could switch from elastic deformation to ploughing, then cutting as the grinding depth increases. With the increase in the grain size, the critical relative grain sharpness (*RGS*) for the transition from elastic deformation to ploughing slightly decreases from 0.23 to 0.16. While the variation of the critical *RGS* for ploughing to cutting transition is more apparent based on the three-dimensional model. When the grain size rises from 4 nm to 12 nm, the critical *RGS* for ploughing to cutting transition decreases from 0.45 to 0.25.

(2) As the grinding depth increases, the dominant subsurface damage mechanism could switch from surface friction into slip motion along the <110> direction. When the grain size increases from 4 nm to 12 nm, less friction-induced damage is generated near the machined surface while more dislocation loops are formed and emitted into the subsurface workpiece. In addition, the proportion of perfect dislocation increases as grain size raises since the larger internal stress is induced in the workpiece. Furthermore, larger residual stress can be observed on the machined surface as the grain size raises.

(3) During the nano-grinding process, the grinding forces rise when the grain radius increases, especially for the normal force. Due to the variation of the dominant grinding mechanism, the average frictional coefficient is decreased when the grain size is enlarged. Meanwhile, an increase in specific grinding energy with increasing grain size is observed, indicating a lower grinding efficiency and, thus, a less energy-efficient grindability when grains with larger size is adopted.

## Figures and Tables

**Figure 1 nanomaterials-13-02670-f001:**
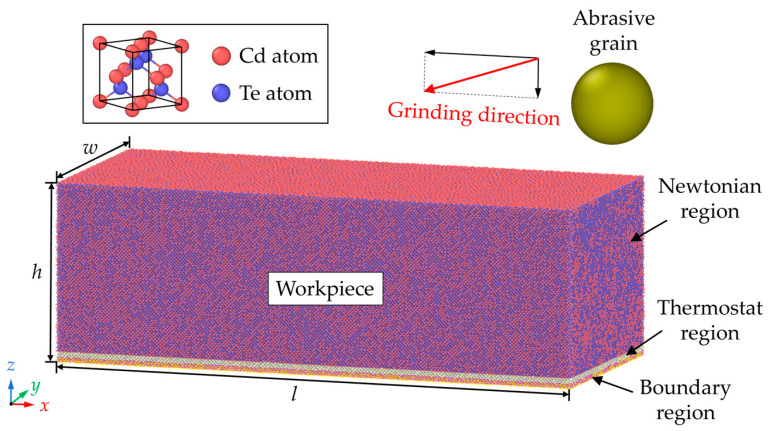
The MD model for nano-grinding simulation of CdTe.

**Figure 2 nanomaterials-13-02670-f002:**
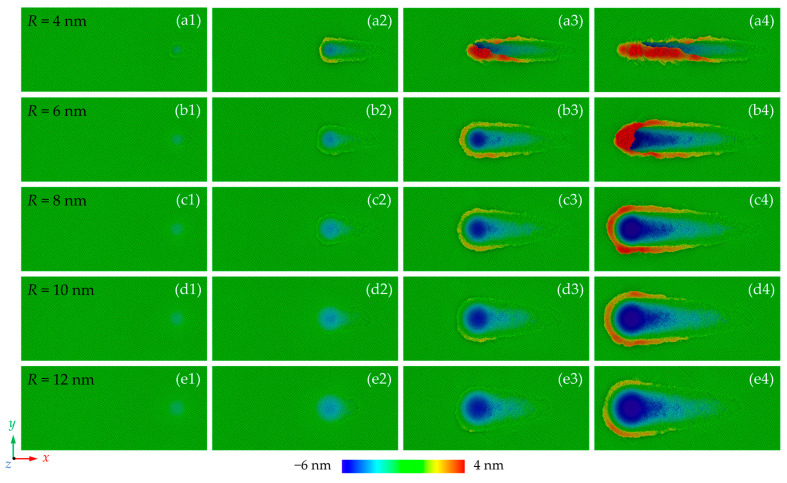
Height distribution of the machined surfaces under different grinding depths and grain sizes: (**a**–**e**) are the surface morphologies with a grain size ranging from 4 nm to 12 nm, while the (1–4) correspond to grinding depth of 1.5 nm, 3 nm, 4.5 nm, and 6 nm, respectively.

**Figure 3 nanomaterials-13-02670-f003:**
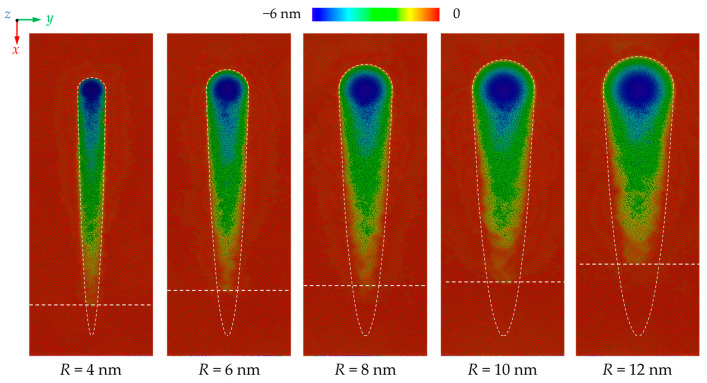
Morphology of the machined groove with different grain sizes where chips and ridges are hidden.

**Figure 4 nanomaterials-13-02670-f004:**
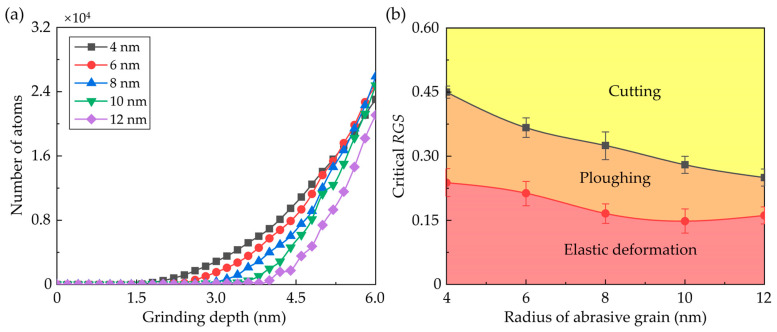
(**a**) Number of the atoms in chips and ridges as a function of grinding depth. (**b**) The critical *RGS* for grinding mechanism transition (Averaged from three simulation cases).

**Figure 5 nanomaterials-13-02670-f005:**
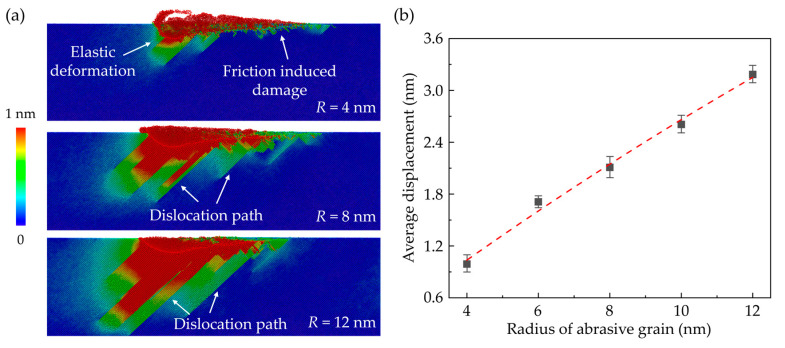
(**a**) Distribution of the atomic displacement magnitudes at a grinding depth of 5 nm when grain radius is 4 nm, 8 nm, and 12 nm, respectively. (**b**) Variation of the average displacement of workpiece atoms as grain radius increases (Averaged from three simulation cases).

**Figure 6 nanomaterials-13-02670-f006:**
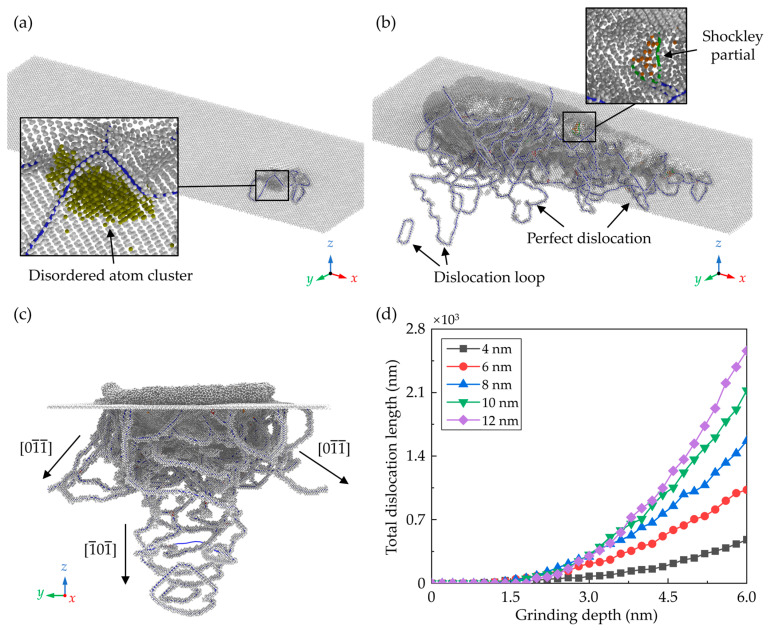
Snapshots of the crystal defects in subsurface at grinding depths of (**a**) 2 nm and (**b**) 6 nm. (**c**) Front view of the crystal defects in subsurface at grinding depth of 6 nm. (**d**) The total length of dislocations during nano-grinding.

**Figure 7 nanomaterials-13-02670-f007:**
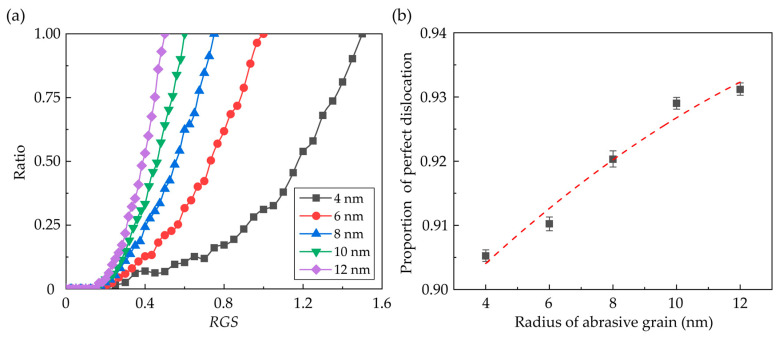
Influence of the grain size on dislocation. (**a**) Ratio of the dislocation length during and after nano-grinding as a function of *RGS*. (**b**) Proportion of perfect dislocation length to the total dislocation (Averaged from three simulation cases).

**Figure 8 nanomaterials-13-02670-f008:**
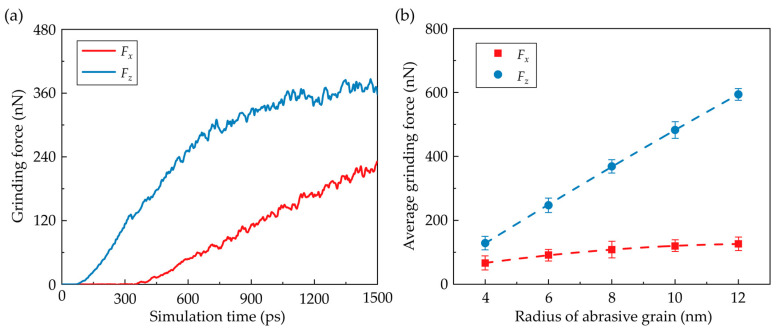
Variation of the grinding forces. (**a**) The transient grinding forces where the grain radius is 6 nm. (**b**) Average grinding forces with different grain sizes (Averaged from three simulation cases).

**Figure 9 nanomaterials-13-02670-f009:**
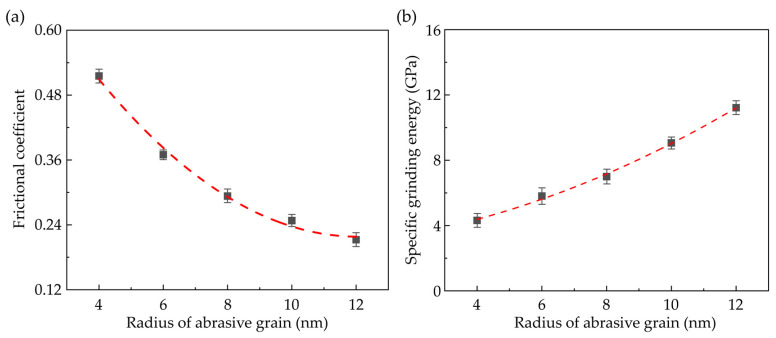
Influence of the grain size on (**a**) the average frictional coefficient and (**b**) specific grinding energy (Averaged from three simulation cases).

**Figure 10 nanomaterials-13-02670-f010:**
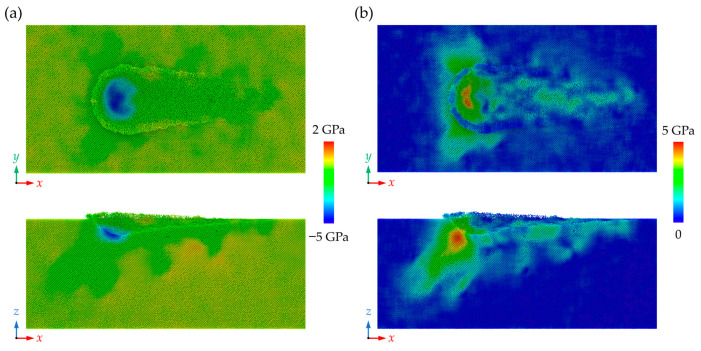
The distribution of the (**a**) hydrostatic stress and (**b**) von Mises stress in workpiece.

**Figure 11 nanomaterials-13-02670-f011:**
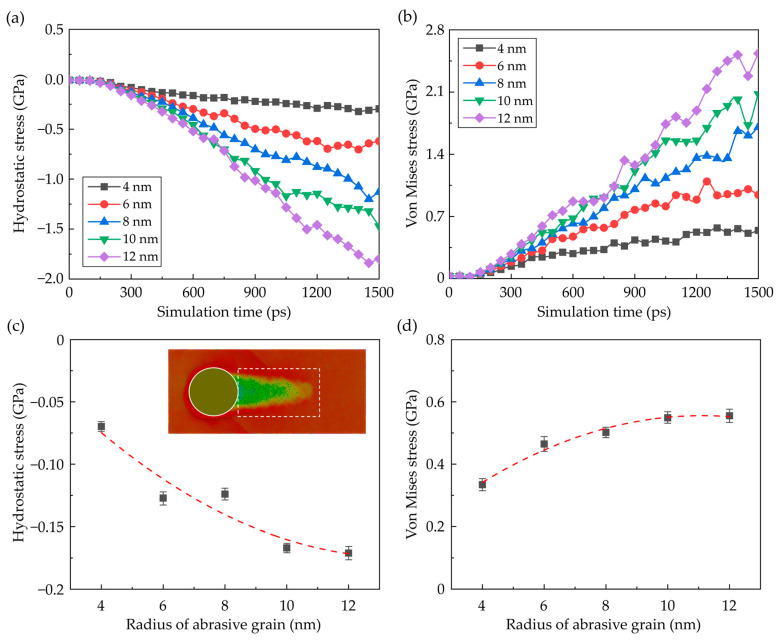
The internal stress in the workplace. (**a**,**b**) The hydrostatic stress and von Mises stress in workpiece during nano-grinding. (**c**,**d**) The average residual stress in the machined surface as a function of grain radius (Averaged from three simulation cases).

**Table 1 nanomaterials-13-02670-t001:** Detailed parameters of the MD model.

Parameters	Value
Workpiece material	Cadmium telluride
Direction of grinding	[$\overset{-}{1}$00] (001)
Size of workpiece (*l* × *w* × *h*)	98 nm × 42 nm × 33 nm
Number of workpiece atoms	About 3.8 million
Radius of the abrasive grain (*R*)	4 nm, 6 nm, 8 nm, 10 nm, 12 nm
Grinding velocity in −*x* direction (*v_x_*)	50 m/s
Grinding velocity in −*z* direction (*v_z_*)	4 m/s
Grinding distance in *x* direction	75 nm
Maximum grinding depth	6 nm
Grinding temperature	300 K

## Data Availability

Not applicable.

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
