# Peer review of "Numerical Investigation on the Effects of Grain Size and Grinding Depth on Nano-Grinding of Cadmium Telluride Using Molecular Dynamics Simulation"

_nanomaterials, 2023, doi:10.3390/nano13192670_

Round 1

Reviewer 1 Report

In this manuscript, MD simulation was conducted to explore the nano-grinding process of cadmium telluride with consideration of the effects of grain size and grinding depth. The authors believe that during nano-grinding, the dominant grinding mechanism could transit from elastic deformation to ploughing, then cutting as the grinding depth increases. The work may be of interest to readers of Nanomaterials. However, the manuscript needs a thorough editing; taking into account the comments below, before I could recommend its publication.

The text of the manuscript should be greatly improved.

1. Introduction

(i) Terms such as "continuum mechanics" and "physical motions" should be avoided in an article on MD simulation.

(ii) Why do authors use classical MD to describe systems in which "... the quantum mechanical effect of materials is dominant"?

(iii) A significant part of the introduction is devoted to describing the results of previous MD simulations of various properties of brittle materials at the nanoscale. At the end of the second paragraph, it is necessary to explain how the approaches described and the results obtained with their help are related to this work.

2. Simulation Methods

This section is described too briefly.

(i) The first derivative of the potential V(r) experiences a discontinuity at the point r = R. Comment!

(ii) Where did the value 5 eV/A3 come from? What happens if it is reduced or increased by 20%?

(iii) The abrasive grain radius varies from 4 to 12 nm. Why are these values taken? The introduction talked about "hundreds of nanometers".

3. Results and discussion

The various curves depicted in Figs. 4, 5(b), 6(d), 7, 8, 9, 11(c), and 11(d) change monotonically with increasing x. On what basis do the authors conclude that: "... the dominant grinding mechanism could transit from elastic deformation to ploughing"?

4. Conclusions.

The first two conclusions are too general, and the third conclusion is obvious. Authors must provide specific values of grain size, at which "... the dominant grinding mechanism could transit from elastic deformation to ploughing". Otherwise, the purpose of MD simulations is not clear.

Reviewer 2 Report

The article is devoted to an important semiconductor material (CdTe) with favorable properties. A compressive understanding of the nanoscale machining mechanism of CdTe is essential for high-quality manufacturing of these materials. Molecular dynamics (MD) simulation is an effective method to explore the deformation and fracture mechanisms of brittle materials at nanoscale. The authors investigated the fundamental deformation mechanisms of CdTe. In particular, the nano-grinding process of CdTe with consideration of the effects of grain size and grinding depth was explored.

The following recommendation can be noted.

On page 3 the authors write: "The nano-grinding simulation is conducted with increasing material removal thickness and various abrasive radii and each case is repeated three times to eliminate the statistical errors". It is good that authors want to elimenate the statistical errors. On the curves in Figures 4-9, confidence intervals should be indicated using the data of three parallel calculations of the authors. This will allow one to understand how significant the difference between the curves on the graphs is.

Round 2

Reviewer 1 Report

The authors took into consideration my suggestions. The manuscript can be recommended for publication.